# Smart Injectable Chitosan Hydrogels Loaded with 5-Fluorouracil for the Treatment of Breast Cancer

**DOI:** 10.3390/pharmaceutics14030661

**Published:** 2022-03-17

**Authors:** Ahmed A. H. Abdellatif, Ahmed M. Mohammed, Imran Saleem, Mansour Alsharidah, Osamah Al Rugaie, Fatma Ahmed, Shaaban K. Osman

**Affiliations:** 1Department of Pharmaceutics, College of Pharmacy, Qassim University, Buraydah 51452, Saudi Arabia; 2Department of Pharmaceutics and Pharmaceutical Technology, Faculty of Pharmacy, Al-Azhar University, Assiut 71524, Egypt; ahmedabdelaal@azhar.edu.eg; 3School of Pharmacy & Biomolecular Sciences, Liverpool John Moores University, James Parsons Building, Liverpool L3 3AF, UK; i.saleem@ljmu.ac.uk; 4Department of Physiology, College of Medicine, Qassim University, Buraydah 51452, Saudi Arabia; malsharidah@qu.edu.sa; 5Department of Basic Medical Sciences, College of Medicine and Medical Sciences, Qassim University, Unaizah 51911, Saudi Arabia; o.alrugaie@qu.edu.sa; 6Department of Zoology, Faculty of Science, Sohag University, Sohag 82524, Egypt; fatmaelzahraa_ahmed@science.sohag.edu.eg

**Keywords:** thermosensitive hydrogel, chitosan, in vitro release, 5-fluorouracil, MTT assay, in vivo antitumor

## Abstract

The treatment of breast cancer requires long chemotherapy management, which is accompanied by severe side effects. Localized delivery of anticancer drugs helps to increase the drug concentration at the site of action and overcome such a problem. In the present study, chitosan hydrogel was prepared for local delivery of 5-Fluorouracil. The in vitro release behavior was investigated and the anticancer activity was evaluated against MCF-7 cells using MTT assay. The in vivo studies were investigated via intra-tumoral injection of a 5-FU loaded hydrogel into breast cancer of female rats. The results indicated that the modified hydrogel has excellent physicochemical properties with a sustained in vitro release profile matching a zero-order kinetic for one month. In addition, the hydrogel showed superior inhibition of cell viability compared with the untreated control group. Moreover, the in vivo studies resulted in antitumor activity with minor side effects. The tumor volume and level of tumor markers in blood were inhibited significantly by applying the hydrogel compared with the untreated control group. In conclusion, the designed injectable hydrogels are potential drug delivery systems for the treatment of breast cancer with a controlled drug release profile, which could be suitable for decreasing the side effects of chemotherapy agents.

## 1. Introduction

Breast cancer is an invasive type of cancer worldwide [1], with 18.2% of all cancer deaths worldwide associated with breast cancer [2]. There are lots of different treatment strategies which have been reported, depending on the type and diagnostic stage of the developed cancer. However, the treatment by chemotherapy remains the most dominant approach [3,4]. As illustrated in Figure 1, 5-Fluorouracil (5-FU) is a very common anticancer drug that can be administered as a single treatment or as adjuvant therapy in many chemotherapeutic protocols. It has been prescribed to treat different types of cancers, including skin, colon, breast, ovarian, and lung cancers [5,6]. The anticancer drug, 5-FU, acts by inhibition of essential biosynthetic processes, or by interfering with macromolecules, like DNA and RNA, and interrupting their normal function [7].

The traditional routes of administration of cytotoxic anticancer drugs reach the whole-body organs, with just a limited amount of drugs reaching the site of the tumor. On the other hand, the nonspecific action of cytotoxic drugs can substantially harm and destroy other healthy organs. [8]. For this reason, there is a need to develop a safe and targeted therapy that achieves high concentration and long-term drug exposure in the tumor tissues with minimal drug levels in the blood and off-target tissue sites. Accordingly, the injectable biodegradable targeted drug delivery systems have been investigated by research scientists [9,10]. Different chemotherapeutic agents, including paclitaxel [11] camptothecin [12], doxorubicin [13], and methotrexate [14] have been prepared in biodegradable polymeric microspheres, hydrogels, surgical pastes, and implants to reduce their cytotoxicity and adverse effects. Thermosensitive hydrogels have a critical solution temperature at which they undergo a transition from a solution phase to a gel phase [15]. Polymers can be injected into the body in a liquid state, followed by gelation at a physiological body temperature to create a cross-linked hydrogel at the optimum solution temperature [16].

As a natural biopolymer derived from chitin by deacetylation, chitosan (Cs) consists of the (1–4) linkage of randomly arranged D-glucosamine and a small proportion of N-acetyl-D-glucosamine (Figure 1) [16,17]. Glycosaminoglycan has been employed in injectable hydrogels due to its antibacterial, biocompatible, biodegradable, and mucoadhesive qualities [18]. The mucoadhesive and in situ Cs gel-forming properties can be used to sustain the delivery of both hydrophilic and hydrophobic drugs [19].

Based on the above, the main objective of this work was the development of a thermosensitive hydrogel from the physical mixture of CS and Pluronic F 127/β-GP for the controlled release of 5-FU. Although this physical combination of polymers has been previously reported by others for intranasal [20], vaginal [21], intravesical [22], ocular [23], and injectable cell delivery carrier for cartilage regeneration [24], and intra-articular [25] delivery, there are no reports of its use for intra-tumoral application in breast cancer. In addition, chitosan/Pluronic gel beads were reported previously [26] and utilized for the release of 5-FU. The results indicated that the drug release was extended for just 24 h. Other groups utilized Cs/PL hydrogels for colon [27] and colorectal [28] delivery of 5-FU. They found that the cumulative release of the drug was achieved within 12 h and 48 h, respectively, depending on the composition of the hydrogel.

In the present study, we focused on the formulation of thermosensitive Cs-hydrogel cross-linked by β-glycerophosphate (β-GP) alone or in combination with 10% Pluronic F127 (PL F127) for localized injection of 5-FU in breast cancer tissues, which to the best of our knowledge, has not been investigated in breast cancer management via loading and controlled release of 5-FU for more than one month. Consequently, the modified hydrogel provides the opportunity for controlled release of the chemotherapeutic agent, which decreases the side effects accompanying the fluctuation in concentration from repeated administration. Practically, a breast tumor model was developed in rats to study the in vivo action of 5-FU loaded Cs-hydrogel on the evolution of the hyperplasia, in comparison with the free aqueous solution of 5-FU. The thermo-reversibility of our hydrogel formula has advantages, including simple preparation and administration via injection without the requirement of operation and anesthesia. The intratumor administration of the drug-loaded gel may help in the concentration of the drug at the site of action and decrease the plasma drug concentration level. Consequently, this will further decrease the side effects. The formulated hydrogel systems were characterized by drug content, pH, and injectability. Moreso, the gel temperature (*T*_gel_) and viscosity were investigated. Both the in vitro release profile and in vitro antitumor activity were carried out in PBS at 37 °C. The in vitro antitumor activity was evaluated against breast cancerous MCF-7 cell membranes via MTT assay. Furthermore, we investigated its toxicological potential via the detection of histopathological appraisals and tumor markers, including CA, CEA, and TNF-alpha. Finally, the modified hydrogels combined the thermo-gelling properties of PL with the properties of biocompatibility, bioadhesiveness, and structural similarity with glycosaminoglycans of chitosan.

## 2. Materials and Methods

### 2.1. Materials

Medical grade chitosan (Mwt, 100–300 kDa and 95% acetylation degree) was provided by Biosyn-tech Inc. (Laval, QC, Canada). Pluronic F127 (Mwt 12,800 Da) was obtained from Fluka Biochemika (Buchs, Switzerland). β-glycerophosphate pentahydrate (β-GP, Mwt 308 Da) were purchased from Loba Chemie PVT LTD. (Mumbai, India) 5-Fluorouracil (5-FU) was obtained from AppliChem (Darmstadt, Germany). DMBA (dimethylbenz (a) anthracene) was provided by Sigma Aldrich Co. (Saint Louis, MO, USA). MCF-7 cells were purchased from Vacsera, Egypt. Acetic acid and dimethylsulphoxide (DMSO) were purchased from El-Nasr Pharm. Chem. Company (Cairo, Egypt). Formalin was provided from Fisher Scientific Limited (Nepean, ON, Canada). DMEM (Dulbecco’s modified Eagle medium) and EDTA (ethylenediaminetetraacetic acid) were purchased from GIB-CO (Carlsbad, CA, USA). All other materials and solvents were of analytical purity and used without further purification.

### 2.2. Preparation and Drug Loading of Chitosan Hydrogel

Semi-synthetic chitosan/Pluronic F127 hydrogel was prepared according to the method by Chenite et al. [29]. Typically, 1.8 g of chitosan was dissolved in 0.1 M acetic acid (100 mL) by stirring for 30 min to obtain a 1.8% *w*/*v* chitosan solution. The aqueous solution of Pluronic F127 (10% *w*/*w*) was mixed with the aqueous solution β-GP (the crosslinking agent) (35% *w*/*w*) to obtain a solution mixture of Pluronic/β-GP which was allowed to be added gradually to the previously prepared chitosan solution at 0 °C for 15 min until the construction of hydrogel. For comparison, a separate hydrogel system was prepared according to the previous procedure but without the addition of Pluronic. The drug loading was achieved by incorporating 5-FU at a concentration of 1% *w*/*v* into the chitosan solution before crosslinking by gentle stirring until complete dissolution. Then, the prepared 5-FU loaded hydrogel systems were stored in the refrigerator at 8 °C until further experiments.

### 2.3. Physicochemical Characterization of the Modified Hydrogel

#### Drug Content, Visual Inspection, and PH Determination

To determine the drug content, the samples of (1 g) hydrogel, loaded with 10 mg 5-FU, were dissolved in distilled water by stirring. The resulting turbid solution was filtered, and the 5-FU concentration was investigated spectrophotometrically at λ_max_ 266 nm [30] using a UV-spectrophotometer (UV-1601, Shimadzu, Kyodo, Japan). In addition, the produced hydrogels’ clarity, pH, phase separation, and homogeneity were visually investigated in either gel or sol forms.

### 2.4. Rheological Studies

An Ostwald U tube capillary viscometer was utilized to determine the viscosity of the produced modified hydrogel systems at 25 ± 1 °C [31]. Furthermore, the viscosity of the modified hydrogel systems (in gel form) was measured at 37 °C and 30 rpm using a Brookfield ultra-viscometer (DV III, RV model with T-bar spindle of T-D 96) (Brookfield Co., Ringgold, GA, USA) [32]. Moreover, the gelation temperature of our system was monitored and recorded by gradual heating at a constant heating rate (1 °C/min) with continuous stirring (30 rpm) using a magnetic bar until the phase transition occurred. The gelation temperature (*T*_gel_) was monitored and reported at a certain temperature, detected once the magnetic bar stopped due to the gelation resistance [33].

### 2.5. Injectability

For any parenteral dosage form, injectability is an important issue. Therefore, the injectability test was implemented by filling a syringe (needle size 5 mm) with the hydrogel mixture (in a liquid form at room temperature) and then allowed to be injected into a meat sample under finger pressure. Importantly, the injectable hydrogel system should have a suitable consistency in order to be extruded through the syringe needle into skin layers for subcutaneous injection [34].

### 2.6. In Vitro Release of 5-FU

Eppendorf tubes were utilized as release cells in which 1 mL of gel containing 1% *w/v* 5-FU were placed and then incubated at 37 °C to ensure the transition of the solution mixture into a hydrogel (sol–gel *T*_gel_ was 29 °C). Then, the loaded gels were covered with 1 mL of PBS pH 7.4 (release medium), and incubated at 37 °C, under shaking at 50 rpm. An amount of 500 µL of the sample was taken at designated interval times and substituted with 500 µL of fresh buffer. The drug concentration in each withdrawn sample was determined spectrophotometrically using a standard calibration curve. The data were expressed as the mean of three separate experiments ± SD.

### 2.7. In Vitro Antitumor Activity (MTT Assay)

The cytotoxicity of the prepared hydrogel systems was investigated against breast cancer cells (MCF-7) using MTT assay [35]. MCF-7 cells were cultivated in DMEM supplemented with 10% fetal bovine serum and 1% *w*/*v* antibiotic combinations of streptomycin (100 µg/mL) and penicillin (100 IU/mL). The MCF-7 was fixed at one million cells/mL and 100 µL were seeded into each well of a 96-well plate and incubated at 37 °C for 24 h. Then, the culture medium was replaced with 0.5 mL of our investigated solutions, which were categorized into different groups: dialyzed chitosan hydrogel solution (negative control), 5-FU^®^ saline solution (positive control), and dialyzed drug-loaded hydrogel solutions (loaded with 5-FU at different concentrations, 200, 300, 500, 700, and 1000 μg/mL). Then, the plate containing the investigated solutions was incubated at 37 °C to create a gel. This was followed by 100 µL fresh medium added to each well and incubated for a further 48 h. Afterward, the incubated cells were treated with MTT reagent (25 µL, 5 mg/mL PBS) and incubated for a further 4 h. After carefully discarding the media, the produced formazan crystals were dissolved using DMSO (200 µL) [36]. After 5 min of shaking, the optical densities were determined at 560 nm using an ELISA microplate reader. Moreover, the effect of the incubation period on cell viability was investigated, taking a fixed concentration (200 µg/mL). The results were presented as the mean of three independent experiments ± SD.

### 2.8. In Vivo Antitumor Studies

#### 2.8.1. Animal Preparation

This in vivo study was approved by the Subcommittee of Health Research Ethics, Deanship of Scientific Research, Qassim University (Approval No: 21-10-07), in accordance with the National Research Council (US) Guide for the Care and Use of Laboratory Animals [37]. Female Sprague Dawley (SD) rats (32 rats) (200.2 ± 15.1 g) were utilized in our current in vivo studies. The rats were delivered from the animal house (Al-Azhar University, Assiut, Egypt). The animals were group-housed in cages (ten rats per cage) in ventilated rooms with a controlled temperature (23 ± 2 °C), and fed with a commercial normal diet and tap water [38]. The induction of breast cancer was carried out via the administration of a cancer-inducing agent, DMBA/olive solution (50 mg/mL). The rats were administered the inducing solution at a concentration of 200 mg/kg body weight with the aid of gavage [39].

The animals were maintained under standard conditions for 8–10 weeks and the six pairs of mammary glands were checked daily by eye inspection, touching, and palpation. The rats were categorized into 4 groups (8 rats each). Group 1 (G1) received no medication and served as a positive control. The second group (G2), which served as a negative control, was injected intratumorally (IT) with a hydrogel solution without drugs. The third group was injected with free 5-FU saline solution at a concentration of 125 mg/kg once monthly. The fourth group was injected with the modified chitosan hydrogel (Cs/p-GP + PL) loaded with 5-FU at a concentration of 125 mg/kg. The change of tumor growth before and after drug injection was monitored and measured periodically. Data were expressed as mean ± SD.

#### 2.8.2. Tumor Volume and Growth Measurement

The first day of tumor appearance was recorded. The mean tumor volume for all animals was examined using a Vernier caliper (micrometer-Ozaki Ltd., Ozaki, Japan) and calculated using Equation (1) [40,41].
TV (mm^3^) = 1/2 × Length × (Width)^2^(1)

The tumor volume was measured, and the treatment was started when the tumor volume reached 500–800 mm^3^ and was considered day 0 (V_0_). Subsequently, the changes in the tumor volume in rats were determined every week after drug administration (Vt). The antitumor effect was monitored by measuring the relative tumor volume (RTV), using Equation (2) [42].
(2)$$\%\;\mathrm{RTV}=\left[\frac{\left(V_{t}\right)}{\left(V_{0}\right)}\right]\times 100$$

V_0_ is the tumor volume directly after drug injection, and Vt represents the tumor volume at each time interval.

#### 2.8.3. Effects on the Lifespan

The antitumor toxicity was also assessed via monitoring the animal’s body weight and the percentage of animal survival after the administration of the anticancer drug. Rats were observed after one month of recovery since the higher percentage of survival was attributed to lower toxicity. However, animal death within the first two weeks after drug administration can be considered a toxic death [43].

#### 2.8.4. Histopathological Appraisals and Traits

After subjecting the investigated animals to inhalation anesthesia, they were sacrificed via cerebral anoxia. Then, tissue samples were taken from the tumor masses of the induced mammary glands and were fixed in 10% formalin at 2–8 °C for 48 h. Afterward, each tumor piece was manually bisected. For the treated groups, the macroscopic localization of the injected site was attempted. The tissues were dehydrated, cleared, and then embedded in blocks of paraffin. Next, the embedded blocks were sectioned into very small sections (5 µm thickness, each). The fixed samples were stained simply by using the usual histological technique (H&E counterstaining of ultrathin paraffin sections (5 µm; Micro HM 360^®^Microtome) [11,44]. The stained sections were examined by a pathologist using a light microscope.

#### 2.8.5. Tumor Markers Detection in Blood

To investigate the tumor’s existence and progress, a blood sample was taken from the animals and subjected to tumor markers investigation. Noteworthy, there are several specific tumor markers for breast cancer, such as carcinoembryonic antigen (CEA) and cancer antigen (CA) [45]. Moreover, an inflammatory mediator, called TNF-α (tumor necrosis factor) was also detected in the blood. The existence of TNF-α at high levels in blood correlates with the proliferation and a higher malignancy grade of breast cancer [46]. The ELISA kit (Anogen, a division of Yes Biotech Laboratories Limited, Mississauga, ON, Canada) was utilized for the detection of all mentioned tumor markers [47].

### 2.9. Statistical Analysis

The comparison and the statistical differences between the investigated groups were determined using the ANOVA one-way program with Duncan’s post hoc test for the paired comparison of means [48]. Statistically significant and very significant differences between data sets were found at *p* < 0.05 and *p* < 0.01, respectively.

## 3. Results and Discussion

### 3.1. Formulation of Thermosensitive Hydrogel Systems

Thermosensitive chitosan hydrogels were constructed successfully with the aid of β-GP and PL F127 as crosslinking agents. The drug loading was carried out during the hydrogel formation. The procedure of hydrogel formation and drug loading ensures the homogenous distribution of the drug which could help in controlling the drug release since the existence of the drug with the chitosan polymer increases its affinity to such a polymer and, consequently, controls the drug release from gel base [49].

### 3.2. Physicochemical Characterization of the Modified Hydrogel System

#### Drug Content, Visual Inspection, and pH Determination

The drug content of the prepared hydrogel was found to be within the range of 98.70 ± 2.85%–102.20 ± 3.80% (Table 1). The produced hydrogel system was evaluated visually before and after gelation. The prepared system was smooth and uniform without any phase separation in both liquid and gel forms. The color was turbid white in the case of the gel form but was clear and colorless in the case of sol form. Moreover, it was observed that the pH of the prepared hydrogel samples was around 6.8 ± 0.2 (Table 1). The pH of injectable hydrogels should be between 6.5 and 7.4 to avoid irritation of tissue [50]. Hence, indicating the suitability of the modified formulations for subcutaneous injection.

### 3.3. Rheological Studies

The viscosity of the modified hydrogel systems, either in sol form or in gel form, is presented in Table 1. The obtained results showed that the viscosity of modified hydrogel in the sol-phase was 1.144 ± 1.5 and 1.429 ± 0.19 Pa/s × 10^−3^, for Cs/β-GP and Cs/β-GP + 10% PL hydrogel system, respectively. In the case of the gel phase, the results showed that the viscosity values were 100.80 ± 1.26 and 120.34 ± 2.34 Pa/s for Cs/β-GP and Cs/β-GP + 10% PL F127 hydrogel system, respectively. The lower viscosity values in the case of sol-phase are suitable for the injection through the needle. Upon injection of the modified hydrogel solution into the body tissues, the temperature of the injected solution will be increased from 25 °C to 37 °C. Consequently, the 3D networks were formed through sol–gel transition (practically achieved at 29 °C, Table 1). This sol–gel transition of our system, including PL copolymer, might be explained by the shift in equilibrium from monomer to micelle, followed by the micellar aggregation to form networks via hydrophobic. Then, the solvent will lose its flowability, and the system converted to gel form [51].

### 3.4. In Vitro Drug Release

The in vitro 5-FU release profiles from two different chitosan hydrogel systems (Cs/β-GP and Cs/β-GP + PL) were carried out in phosphate buffer pH 7.4 at body temperature (37 °C). The experiments were allowed to continue until the quantitative release of all the loaded drug to exclude the probability of a drug-polymer interaction or drug degradation during the long period of measurements.

The results, illustrated in Figure 2, showed that the release rate of 5-FU from system (A) hydrogel (composed of Cs and β-GP) was relatively faster since the quantitative drug release was achieved after two weeks, compared with that obtained in the case of system (B) hydrogel (composed of Cs/β-GP + PL), in which the quantitative drug release was achieved after one month. The obtained results can be attributed to the existence of PL polymer which increased the crosslinking and, consequently, the viscosity of the final hydrogel system, as indicated in Table 1. The obtained results are in good agreement with previously obtained data since it provides a more sustained release profile with higher viscosity gel systems [52].

Interestingly, the more sustained and controlled release of anticancer drugs, as shown in our formulation, is more suitable than a shorter release rate to attain the steady-state concentration and decrease the fluctuation of plasma concentration and the accompanying side effects [14].

### 3.5. Injectability

The ability of the modified hydrogel systems to be injected easily from 5 mm syringes was investigated. It was observed that in the case of our modified hydrogel system B (Cs/β-GP + PL), a volume of 0.5 mL was extruded through the syringe needle within 10 s upon pressing by fingers. This finding indicated that the injecting of these solutions from the syringes to the muscles was very simple (no operation is needed) which comes with patient compliance [53].

### 3.6. In Vitro Antitumor Activity

As shown in Figure 3, all medicated groups (either free 5-FU or 5-FU -loaded Cs/β-CD + PL hydrogel showed a significant (*p* ˂ 0.05) higher tumor cell viability inhibition when compared with either blank or untreated groups. Interestingly, the non-loaded hydrogel had a little inhibitory effect on tumor cells, which was attributed to the interaction of positively charged Cs polymer with the negatively charged cancerous cells [54,55,56,57]. Moreso, it was observed that the longer the treatment time, the more extensive the inhibition effect. In the case of free 5-FU, the extent of cell viability inhibition was 70%, 52%, and 34.25% cell viability for 24, 48, and 72 h incubation periods, respectively. Whereas, in the case of 5-FU loaded Cs/β-GP + PL hydrogel, the cell viability values were 80%, 50%, and 20.6% for 24, 48, and 72 h incubation periods, respectively. The results showed that the loaded Cs/β-GP + PL hydrogel was found to have a significant (*p* ˂ 0.05) anti-proliferation capacity, especially for MCF-7 cells at 72 h with 20.61% cell viability compared with 34.25% viability for a free 5-FU solution. These results indicated that the drug-loaded gel has an extended duration of action. In contrast, at the 24 h incubation period, it was observed that 5-FU solution had a higher inhibitory impact on cell viability compared with the drug-loaded gel group. This may be due to the partial dilution effect of non-toxic Cs polymer existing within the composition of gel.

The effect of concentration on cell viability at the incubation period of 48 h was investigated, and the data was graphically presented in Figure 4. The results indicated that the inhibitory extent on cells is concentration-dependent, since the higher the solution concentration the higher the extent of reduction on cell viability. The IC_50%_ for all groups was observed and recorded from the obtained Figure 3 and Figure 4 during the experiment. It was observed that the values of IC_50%_ were 0.19 and 0.22 µg/mL for 5-FU solution and 5-FU-loaded Cs/β-GP + PL hydrogel. respectively. Moreso, the IC_50%_ was achieved within 48 h of incubation.

### 3.7. In Vivo Antitumor Activity

#### 3.7.1. Effect of Cytotoxicity on Relative Tumor Volume

The antitumor effect of the prepared hydrogel system was compared with that of 5-FU by measuring the relative tumor volume of rats after tumor induction using DMBA administration. The first tumor was observed eight weeks after DMBA administration. By the end of the ten weeks’ study, all rats developed at least one breast tumor. The mean tumor volumes, calculated from the caliper, were 652.5 ± 7.05 mm^3^. Regarding the antitumor activity of different groups, the results showed that there was a significant (*p* ˂ 0.05) difference in the tumor growth between the negative control (unmedicated hydrogel solution) and all other groups (either treated with 5-FU^®^ saline solution or with 5-FU-loaded Cs/β-GP + PL hydrogel). During a period of three weeks, we monitored the changes in relative tumor volume in rats and the data was presented in Figure 5. It was observed that 5-FU (either in saline solution or in the form of a loaded hydrogel) suppressed significantly (*p* ˂ 0.05) the tumor growth when compared with control groups. Nonetheless, it was found that the 5-FU Cs/β-GP + PL hydrogel system had a significantly (*p* < 0.05) higher inhibition of RTV compared with the 5-FU saline solution.

Interestingly, the group treated with non-loaded Cs/β-GP + PL showed a partial tumor growth inhibition compared with the untreated control group but was not significant. This finding may be attributed to the ability of chitosan polymer to penetrate cancer cell membranes bearing negative charges [56,57]. This result confirms the previously obtained data of cell viability studies (Figure 3 and Figure 4). For the same reason, the hydrogel-loaded 5-FU showed a significant growth inhibition (i.e., lower tumor volume, 12% RTV) compared with free drug (i.e., higher tumor volume, 40% RTV) after 7 days. The activity of the modified gel was extended for more than two weeks (8% RTV) but the free drug activity was diminished by time (60% RTV) due to its short duration of action. The obtained results and interpretation were in good agreement with the reported data [56,57].

#### 3.7.2. Cytotoxic Impact on Rat’s Body Weight and Mortality

The animal’s weight and survival rates of treated animals were monitored for an additional four weeks after treatment to follow up on the toxic side effect of chemotherapeutic drugs. Generally, the results, as illustrated in Figure 6, showed a significant (*p* < 0.05) weight loss in the control untreated group compared with the 5-FU-loaded Cs/β-GP + PL hydrogel receiving group. These results may be due to alleviating the side effects of free drugs through sustained release [58]. Especially, by the end of week three, it was observed that there was a significant difference between the groups (*p* < 0.05) from the third week to the end of the experiment since it was observed that the body weight of the investigated rats was around 190 g in the case of both normal and treated groups. However, the body weight decreased with time to 173, 165, and 159 g at weeks 4, 6, and 8, respectively. The results, illustrated in Figure 6, showed that the rats receiving 5-FU displayed weight loss throughout the first 7–10 days of the experiments. In contrast, the rats receiving 5-FU-loaded hydrogel had weight curves nearly similar to those of the ordinary animals (-ve control group).

Regarding the degree of mortality, the results showed that none of the rats treated with the prepared hydrogel system died during the experimental period compared with three rats that died in the positive control group.

#### 3.7.3. Histopathological Appraisals and Traits

Tumor-induced samples showed marked histopathological changes in both the mammary gland tissue and its covering skin. Large proliferating neoplastic cells with marked dysplasia were detected in the acini and ducts of the mammary glands. The neoplastic carcinoma cells grouped and formed clusters with a low tendency for acini formation. The proliferating neoplastic cells formed multiple layers, papillary projections inside the mammary ducts. Malignancy in the form of frequent atypical mitosis, increasing nuclear/cytoplasmic ratio, hyperchromatic nuclei, and cellular and nuclear pleomorphism was observed on the carcinoma cells. In addition, marked inflammatory edema with intense inflammatory cells infiltration was seen in the surrounding tissues. Notably, the skin that covers the affected glands showed acanthosis with marked vacuolation in the prickle cell layer. Its surface was covered by a thick zero cellular crust with intense inflammatory cells infiltration. Some of the severely affected cases showed squamous cell carcinoma in the skin covering the gland with the existence of the characteristic ‘epithelial pearl’ that was made up of central keratin whorl surrounded by neoplastic epithelial cells. In chitosan-treated samples, marked improvement was observed, as the detected tumor cells were less frequent and had a higher tendency for acini and duct formation without inward growths, cells clusters, or sheets, and minimal inflammatory reactions were observed (Figure 7).

#### 3.7.4. Tumor Markers Detection in Blood

The serological examination of breast cancer in the blood, both CEA (Figure 8A) and CA (Figure 8B), were quantitatively determined. The CA marker was measured to determine the presence of breast cancer, while CEA indicates the progression of cancer to other tissue or organs. Figure 8A–C showed the recorded tumor parameters for all groups over two weeks of treatment. The results indicated that both the investigated parameters were high in control and blank Cs–Gel, which showed a 14-fold higher level of tumor markers than the normal (tumor-free group). The values of both the two tumor parameters were reduced significantly (*p* ˂ 0.05) in all treated groups, with the highest extent of reduction obtained in the case of 5-FU loaded Cs/β-GP + PL. Similarly, it was found that the level of TNF-α (Figure 8C) was significantly elevated (490 pg/mL ) in the case of the control group that received no medication, compared with normal rats (25 pg/mL). In contrast, its level was reduced significantly (*p* ˂ 0.05) by the injection of antitumor since 5-FU loaded Cs–Gel showed the highest effect (45 pg/mL) compared to free 5-FU treated groups (32 pg/mL). The obtained data are in agreement with the obtained histopathological data (discussed above) which showed a reduction in swelling and inflammation accompanying the tumor proliferation.

## 4. Conclusions

In this study, 5-FU loaded into the semi-synthetic hydrogel system was successfully constructed from chitosan and a β-GP and PL copolymer. The prepared hydrogel system showed an extended and controlled drug release (30 days) with good physicochemical properties. The in vitro MTT assay indicated the drug-loaded hydrogel system can kill tumor cells within 72 h depending on the concentration and incubation period. Moreover, the in vivo studies showed the ability of our system to not only inhibit the tumor growth but also decrease its size to the normal stage. The obtained results were confirmed by both histopathological manifestations and the reduction in tumor markers in the blood. The obtained results suggested that the localized injection of the modified hydrogel system could significantly inhibit the tumor growth, on one hand, and avoid expected drug adverse effects. As a result, it may be recognized as a promising and efficient approach for achieving the sustained quantitative release of anticancer medications.

## Figures and Tables

**Figure 1 pharmaceutics-14-00661-f001:**
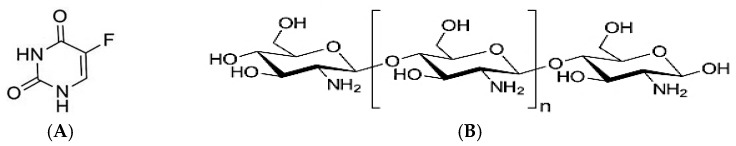
Chemical structure of 5-FU (**A**) and chitosan (**B**).

**Figure 2 pharmaceutics-14-00661-f002:**
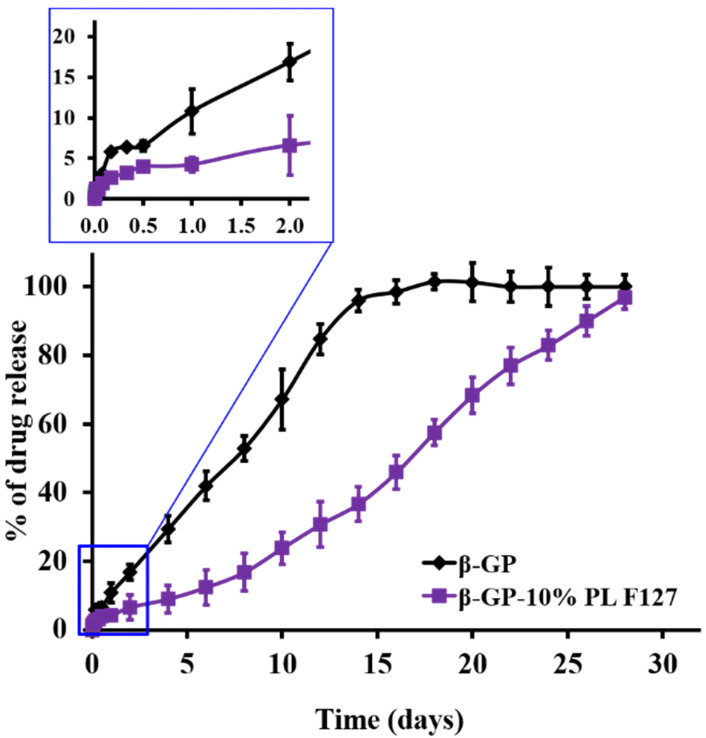
The in vitro release profiles of 5-FU 1% (*w*/*v*) form either hydrogel system A (chitosan/β-GP) or hydrogel system B (chitosan/β-GP + PL) at 37 °C. The results are the mean of three independent experiments ± SD.

**Figure 3 pharmaceutics-14-00661-f003:**
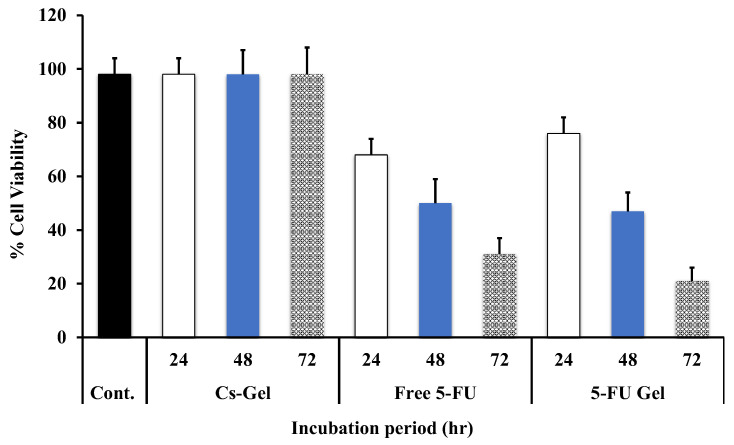
Cytotoxicity studies of 5-FU-loaded Cs/β-GP + PL hydrogel on MCF-7 cells in comparison with either blank chitosan hydrogel or free drug solution, as a function of the incubation period (concentration was fixed at 200 µg/mL). The obtained results are shown as the mean of three experiments ± SD.

**Figure 4 pharmaceutics-14-00661-f004:**
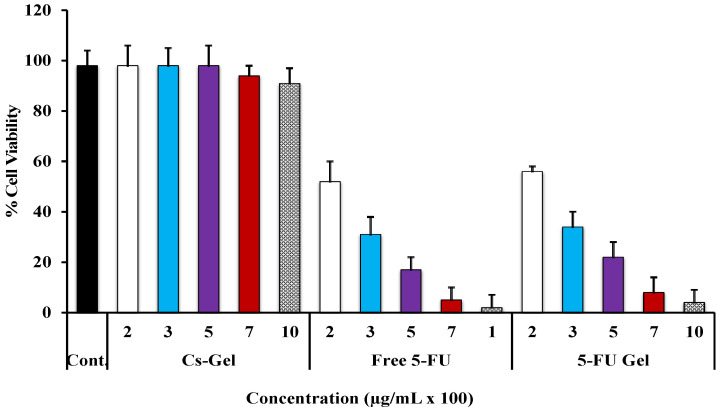
Cytotoxicity studies of 5-FU-loaded Cs/β-GP + PL hydrogel on MCF-7 cells in comparison with either blank chitosan polymer or free drug solution, as a function of concentration (incubation period was fixed at 48 h). The obtained results are shown as the mean of three experiments ± SD.

**Figure 5 pharmaceutics-14-00661-f005:**
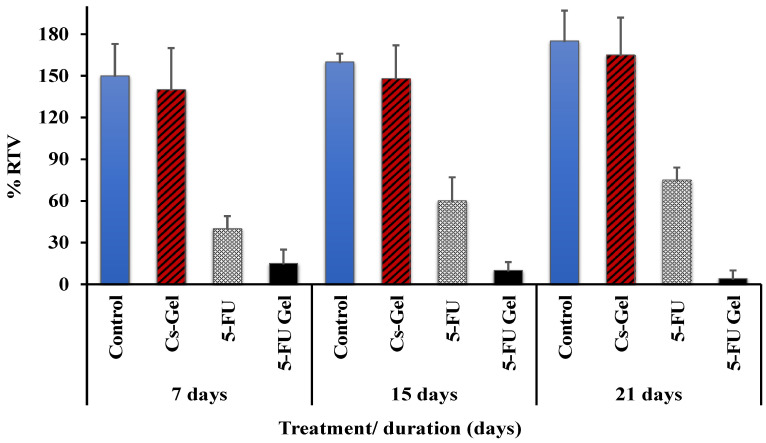
The antitumor activity (RTV; the changes of relative tumor volume) of either 5-FU saline solution or 5-FU-loaded Cs/β-GP + PL after injection into the breast tumor. (*n* = 8).

**Figure 6 pharmaceutics-14-00661-f006:**
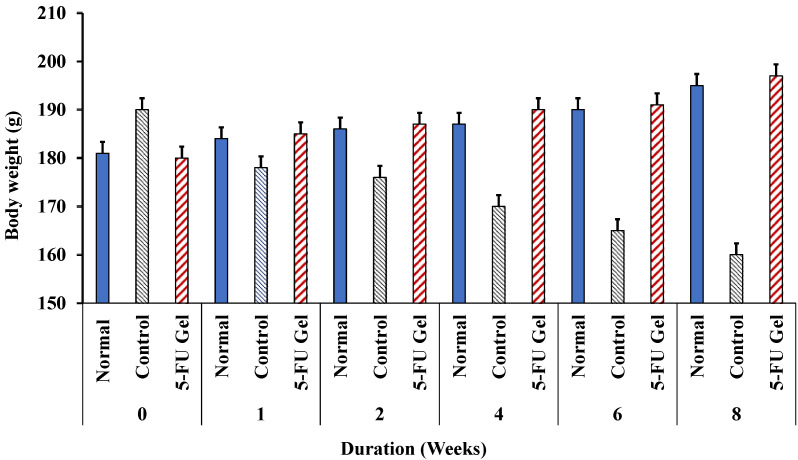
The relative body weight changes of normal rats (have no tumor), control rats (untreated tumor-bearing rats) and rats treated with 5-FU loaded Cs/β-GP + PL gel. The results are presented as the mean ± SD (*n* = 8).

**Figure 7 pharmaceutics-14-00661-f007:**
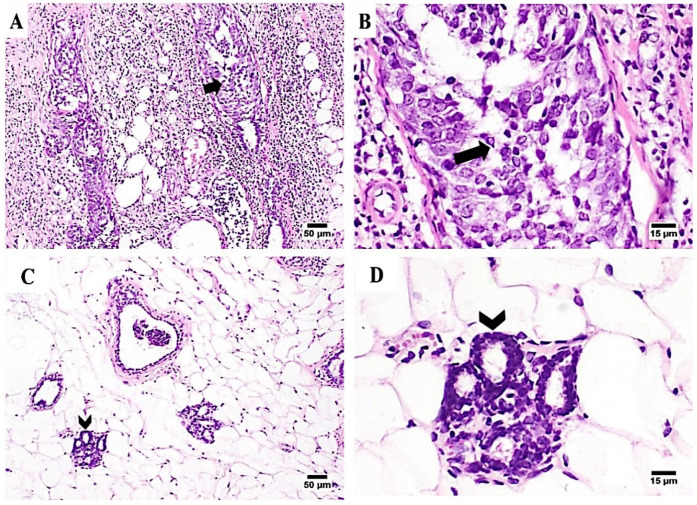
Photomicrographs showing the histopathological traits observed on tumor-induced mammary gland tissues. (**A**,**B**) Non-treated mammary gland showing malignant cells proliferation (arrows); H&E 100×, and 400×, respectively. (**C**,**D**) Mammary gland treated with chitosan hydrogel showing proliferating neoplastic cells forming ducts, without inward growths, cells clusters, or sheets, with mild inflammatory cells infiltration (chevrons); H&E 100×, and 400×, respectively.

**Figure 8 pharmaceutics-14-00661-f008:**
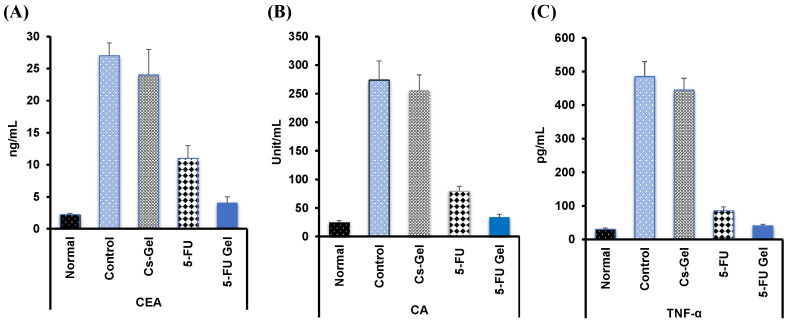
Effect of injected 5-FU hydrogel systems on breast tumor markers (CEA (**A**), CA (**B**), TNF-alpha (**C**)) which were determined in the blood of the investigated groups of rats. The data were presented as the mean ± SD (*n* = 3).

**Table 1 pharmaceutics-14-00661-t001:** Physicochemical properties of the modified hydrogel systems.

Gel Composition	Drug Content	PH	Viscosity (Pa/s)	Gel Point (°C)
Gel (A): (Cs/β-GP)	102.20 ± 3.80	6.7 ± 0.3	100.80 ± 1.26	29.3 ± 1.5
Gel (B): (Cs/β-GP + PL)	98.70 ± 2.85	6.8 ± 0.2	120.34 ± 2.34	29.3 ± 1.5

Cs: chitosan, β-GP: beta-glycerophosphate and PL: Pluronic F127.

## Data Availability

Not applicable.

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
