# Peer review of "Smart Injectable Chitosan Hydrogels Loaded with 5-Fluorouracil for the Treatment of Breast Cancer"

_pharmaceutics, 2022, doi:10.3390/pharmaceutics14030661_

Round 1

Reviewer 1 Report

It’s a very interesting paper on novel potential hydrogel anti-cancer drug delivery systems.

My detailed remarks are the following:

  1. The data of drug released should be more discussed. The kinetics and mechanism of in vitro 5-FU release should be characterized by zero order, first order, Higuchi and Korsmeyer–Peppas mathematical models.
  2. How was the stability of obtained hydrogel drug delivery systems?

Author Response

Feb 28, 2022

Editor-in-Chief

Pharmaceutics

Dear Editor:

We wish to re-submit the manuscript titled “Smart injectable chitosan hydrogels loaded with 5-fluorouracil for the treatment of breast cancer”. The manuscript ID is pharmaceutics-1613172.

We thank you and the reviewers for your thoughtful comments on the manuscript and have edited it to address these concerns. We appreciate your insightful suggestions which have helped to improve the manuscript. The responses to all comments have been prepared and attached herewith. We have also ensured that our revised files adhere to the author guidelines of your esteemed journal.

Reviewer #1

It’s a very interesting paper on novel potential hydrogel anti-cancer drug delivery systems.

My detailed remarks are the following:

Questions

Responses

Line No

1. The data of drugs released should be more discussed. The kinetics and mechanism of in vitro 5-FU release should be characterized by zero-order, first-order, Higuchi, and Korsmeyer–Peppas mathematical models.

Thanks a lot for the comment. Indeed, the kinetics of release profiles were investigated and the results showed that the release mechanism was zero order in the case of both two modified formulas. The data was not shown but was referred to in the text due to the manuscript is full of other data.

2. How was the stability of obtained hydrogel drug delivery systems?

The stability study with other analyses will be the issue of our next manuscript

We look forward to working with you and the reviewers to move this manuscript closer to publication in J of Pharmaceutics.

Sincerely,

Shaaban K. Osman

Ahmed A. H. Abdellatif

Reviewer 2 Report

  1. The abstract exceeds the maximum number of words.
  2. The Introduction presents general information and does not present an analysis of the current state of what is developed in this manuscript. Therefore, the authors do not show us the novelty of the study undertaken.
  3. For reference 20, the first author should be mentioned in the text.
  4. The same font must be used in the entire manuscript.
  5. Section 2.6, does not clearly describe the method. How was performed the dilution, since a very small sample quantity was extracted for spectrophotometric analysis?
  6. In section 3.4, how do the authors appreciate the release of 5-FU of 100% of the β-GP product? Why does this product present this behavior in comparison with other products? As evidenced by the explanation presented in the article, the phenomenon is due to crosslinking. Did the authors determine the degree of crosslinking?
  7. How is demonstrated that the 5FU is successfully loaded in the hydrogel? Moreover, regarding the release of 5-FU, how the quantification of these was done. No calibration curve is performed, which makes me doubt the results.
  8. Reference 32: Habiba, F .; Hassana, M .; Sayeh, F .; Ela, A .; Raheem, R. Different topical formulations of ketorolac 562 tromethamine for anti-inflammatory application and clinical efficacy. Dig J Nanomater Biostruct. 9, 563 (2014), 705-719 - is not correct, the last “a” is added from the affiliation to the names of the first 2 authors, it is correct (Habib, F .; Hassan, M.
  9. Equation 2 is repeated, it is the same as equation 3, where it should be relative tumor inhibition rate (TIR,%).
  10. In lines 332-337, the authors present higher experimental results compared to the results shown in Figure 3. A clear example is a value of 40% for free 5-FU at 72 h, then in line 336, it is clear that he obtained only 34.25%, as can be seen from the graph.
  11. In Figure 4, for Free 5-FU, the value 10 must be instead of 1.
  12. At line 359, the study activity should be modified according to the experimental part.
  13. Figure 7 is missing from the text.
  14. Bibliographic references do not follow the same format.

Author Response

Feb 28, 2022

Editor-in-Chief

Pharmaceutics

Dear Editor:

We wish to re-submit the manuscript titled “Smart injectable chitosan hydrogels loaded with 5-fluorouracil for the treatment of breast cancer”. The manuscript ID is pharmaceutics-1613172.

We thank you and the reviewers for your thoughtful comments on the manuscript and have edited it to address these concerns. We appreciate your insightful suggestions which have helped to improve the manuscript. The responses to all comments have been prepared and attached herewith. We have also ensured that our revised files adhere to the author guidelines of your esteemed journal.

Reviewer #2

Comments and Suggestions for Authors

Questions

Responses

Line No

1. The abstract exceeds the maximum number of words.

Thanks for the hint. It was adjusted properly and reduced to 200 words.

Abstract section

2. The Introduction presents general information and does not present an analysis of the current state of what is developed in this manuscript. Therefore, the authors do not show us the novelty of the study undertaken.

The novelty of our manuscript was focusing and highlighting the application of the hydrogel system in loading and localized intratumor injection of 5-FU for breast cancer management, using different tumor analysis techniques.

3. For reference 20, the first author should be mentioned in the text.

Reference 20 was updated as the reviewer suggested

References section

4. The same font must be used in the entire manuscript.

Thanks a lot for the hint. It was adjusted.

5. Section 2.6, does not clearly describe the method. How was performed the dilution, since a very small sample quantity was extracted for spectrophotometric analysis?

Thanks for the comment. However, indeed, the released samples (500µl) represent half of the release medium volume (1 ml). So, it was concentrated with a drug which give us the chance for dilution with PBS buffer (500µl) to the extent which can be measured at the specific wavelength.

see section 2.6

6. In section 3.4, how do the authors appreciate the release of 5-FU of 100% of the β-GP product? Why does this product present this behavior in comparison with other products? As evidenced by the explanation presented in the article, the phenomenon is due to crosslinking. Did the authors determine the degree of crosslinking?

Thanks for the comment. In the present work, we evaluated the extent of cross-linking by measuring the viscosity of the produced hydrogel. We found that the hydrogel formula composed of pluronic is higher than that obtained in the case of the formula without pluronic.

It was highlighted in the text.

Line 278-282

7. How is demonstrated that the 5FU is successfully loaded in the hydrogel? Moreover, regarding the release of 5-FU, how the quantification of these was done. No calibration curve is performed, which makes me doubt the results.

5-FU is a soluble compound. It was loaded during the hydrogel formation by simple mixing.

Regarding the quantification of the released drug, the calibration curve was carried out, for sure, without such a standard curve we can not convert the absorbance to concentration

Line

107-109

And line 144

8. Reference 32: Habiba, F .; Hassana, M .; Sayeh, F .; Ela, A .; Raheem, R. Different topical formulations of ketorolac 562 tromethamine for anti-inflammatory application and clinical efficacy. Dig J Nanomater Biostruct. 9, 563 (2014), 705-719 - is not correct, the last “a” is added from the affiliation to the names of the first 2 authors, it is correct (Habib, F .; Hassan, M.

The reference was updated as the reviewer suggested

9. Equation 2 is repeated, it is the same as equation 3, where it should be relative tumor inhibition rate (TIR,%).

Thanks for the hint. It was corrected

Line 199

10. In lines 332-337, the authors present higher experimental results compared to the results shown in Figure 3. A clear example is a value of 40% for free 5-FU at 72 h, then in line 336, it is clear that he obtained only 34.25%, as can be seen from the graph.

It was corrected properly

Lines

312, 317

11. In Figure 4, for Free 5-FU, the value 10 must be instead of 1.

Thanks. It was corrected in the figure.

Lines

336, 339

12. At line 359, the study activity should be modified according to the experimental part.

Thanks for the comment. It was modified properly.

Lines

189-207

13. Figure 7 is missing from the text.

Thanks a lot for the hint. It was introduced at a suitable site in the text.

Lines

411, 412

14. Bibliographic references do not follow the same format.

The bibliographic references were updated using endnote 9.

We look forward to working with you and the reviewers to move this manuscript closer to publication in J of Pharmaceutics.

Sincerely,

Shaaban K. Osman

Ahmed A. H. Abdellatif

Reviewer 3 Report

The authors have made an interesting article describing the development of Smart injectable chitosan hydrogels loaded with 5-fluorouracil for the treatment of breast cancer but some improvements should be made:

  1. The writing formation is not uniform done, there are some characters written in a different size (such as Line 27-28, 61-65, 87, 103-104, 119-120, 221 etc)….please revise all the article according to the journal’s instructions.
  2. At line 106, the phrase should start with: “All other materials …. “ line 161 “100uL each” should be replaced with “100µl each”.
  3. The following phrases from 169-170 lines should be reformulated: “Afterwards, the incubated cells were treated with MTT reagent (25 µL, 5 mg/mL PBS) and allowed to be for 4h further. after carful discarding of the media, DMSO (200 µL) was added in order 170 to solubilize the produced formazan crystals”.
  4. Line 186 the proposition should start with capital letter. The same line 220.
  5. The following phrase from line 218 should be finished: “After subjecting of the investigated animals to inhalation anesthesia, they were sacrificed via cerebral”. Cerebral what? Dislocation?
  6. Line 284 erase the dot before the reference number.
  7. Table 1 should be written uniform, the spaces between ± should be the same each time a value is written and also be consistent in the representation of a value regarding the number of decimals used.
  8. When referring in the text to a table, the t should be written with capital letter.
  9. At line 312-314 it is stated that “after a volume of 0.5 ml was extruded through the syringe needle within 10 seconds upon pressing by fingers indicates that the injecting of these solutions from the syringes to the muscles was painless”. How do the authors reached to the conclusion that injecting these solutions is painless??
  10. Line 319 “determined d” please erase the d. In line 322 the concentration aren’t written correctly 200.300, 500, 700 and 1000 μg/mL.
  11. Line 328 What this formulation stands for: “This minute inhibitory”? Please be more precise and detailed in expression.
  12. The reference on the figure in text is not done in a uniform way, sometimes is bold sometimes is regular written and sometimes is written with capital letter.
  13. How the IC50% was determined? Please complete this information.
  14. Line 354 Proposition should start with capital letter.
  15. Line 383 Please explain the phrase: “the hydrogel loaded 5-FU showed a significant growth inhibition (12% RTV) compared with free drug (40% RTV)”. The inhibition of tumor growth was only 12%???

Author Response

Feb 28, 2022

Editor-in-Chief

Pharmaceutics

Dear Editor:

We wish to re-submit the manuscript titled “Smart injectable chitosan hydrogels loaded with 5-fluorouracil for the treatment of breast cancer”. The manuscript ID is pharmaceutics-1613172.

We thank you and the reviewers for your thoughtful comments on the manuscript and have edited it to address these concerns. We appreciate your insightful suggestions which have helped to improve the manuscript. The responses to all comments have been prepared and attached herewith. We have also ensured that our revised files adhere to the author guidelines of your esteemed journal.

Reviewer #3

The authors have made an interesting article describing the development of Smart injectable chitosan hydrogels loaded with 5-fluorouracil for the treatment of breast cancer but some improvements should be made:

Questions

Responses

Line No

1. The writing formation is not uniform done, there are some characters written in a different size (such as Line 27-28, 61-65, 87, 103-104, 119-120, 221 etc)….please revise all the article according to the journal’s instructions.

Thanks a lot for the comment. It was readjusted properly.

2. At line 106, the phrase should start with: “All other materials …. “ line 161 “100uL each” should be replaced with “100µl each”.

Thanks for the hints, they were corrected and highlighted in the text

Lines

97, 107-109

151, 160-162

3. The following phrases from 169-170 lines should be reformulated: “Afterwards, the incubated cells were treated with MTT reagent (25 µL, 5 mg/mL PBS) and allowed to be for 4h further. after carful discarding of the media, DMSO (200 µL) was added in order 170 to solubilize the produced formazan crystals”.

It was rephrased and highlighted

Lines

159-162

4. Line 186 the proposition should start with a capital letter. The same line 220.

The manuscript was modified as suggested by the reviewer.

177,

220

5. The following phrase from line 218 should be finished: “After subjecting of the investigated animals to inhalation anesthesia, they were sacrificed via cerebral”. Cerebral what? Dislocation?

It was rephrased and highlighted

6. Line 284 erase the dot before the reference number.

The manuscript was modified as suggested by the reviewer.

268

7. Table 1 should be written in uniform, the spaces between ± should be the same each time a value is written, and also be consistent in the representation of a value regarding the number of decimals used.

Thanks a lot for the hint. It was rewritten properly.

Line 266

8. When referring in the text to a table, the t should be written with capital letters.

Thanks a lot. It was done.

Lines

254, 262

9. At lines 312-314 it is stated that “after a volume of 0.5 ml was extruded through the syringe needle within 10 seconds upon pressing by fingers indicates that the injecting of these solutions from the syringes to the muscles was painless”. How do the authors reach the conclusion that injecting these solutions is painless??

Thanks for the comment. It was rephrased and highlighted in the text.

Line 296

10. Line 319 “determined d” please erase the d. In line 322 the concentration isn’t written correctly 200.300, 500, 700, and 1000 μg/mL.

They were corrected in the text.

Line 300, 303

11. Line 328 What this formulation stands for: “This minute inhibitory”? Please be more precise and detailed in expression.

The whole sentence was rephrased.

Lines

308-310

12.The reference on the figure in text is not done in a uniform way, sometimes is bold sometimes is regular written and sometimes is written with capital letter.

They were adjusted uniformly within the whole text.

Lines

306, 317, 332, 337

13.How the IC50% was determined? Please complete this information.

It was determined via recording the time or concentration at which the number of cells reduced to its half.

Line 327

14.Line 354 Proposition should start with capital letter.

Done.

Line 340

Line 383 Please explain the phrase: “the hydrogel loaded 5-FU showed a significant growth inhibition (12% RTV) compared with free drug (40% RTV)”. The inhibition of tumor growth was only 12%???

There is an indirect relationship between the tumor volume (RTV) and the % inhibition.

The higher the inhibition % the lower the tumor volume. It was explained in the text.

Lines 365-367

We look forward to working with you and the reviewers to move this manuscript closer to publication in J of Pharmaceutics.

Sincerely,

Shaaban Osman

Ahmed A. H. Abdellatif

Round 2

Reviewer 2 Report

The authors have made an effort to improve the quality of the manuscript, but I still request the authors and support the need to demonstrate what is new in this study in relation to the current state of the art. The introduction of the manuscript must be completed in accordance with this issue.

Furthermore, the bibliography is not uniformly edited.

Author Response

Mar 4, 2022

Editor-in-Chief

Pharmaceutics

Dear Editor:

We wish to re-submit the manuscript titled “Smart injectable chitosan hydrogels loaded with 5-fluorouracil for the treatment of breast cancer”. The manuscript ID is pharmaceutics-1613172.

We acknowledge the reviewers for the thoughtful comments on the manuscript. The responses to all comments have been prepared and attached herewith. We have also ensured that our revised files adhere to the author guidelines of your esteemed journal.

Reviewer 2

1- The authors have made an effort to improve the quality of the manuscript, but I still request the authors and support the need to demonstrate what is new in this study in relation to the current state of the art. The introduction of the manuscript must be completed in accordance with this issue.

Response;

Thanks a lot for the effort and the valuable comment. Accordingly, the introduction has been improved by the addition of new paragraphs describing the novelty of our work, in line 71.

2- Furthermore, the bibliography is not uniformly edited.

Response;

The bibliography was updated using endnote 9, as per instruction of mdpi website (Author information).

https://www.mdpi.com/authors/references

We look forward to working with you and the reviewers to move this manuscript closer to publication in J of Pharmaceutics.

Sincerely,

Shaaban K. Osman

Ahmed A. H. Abdellatif

Reviewer 3 Report

The authors made the proper modification according to the indications and it it now ready to be submitted to the journal for publication.

Author Response

Mar 4, 2022

Editor-in-Chief

Pharmaceutics

Dear Editor:

We wish to re-submit the manuscript titled “Smart injectable chitosan hydrogels loaded with 5-fluorouracil for the treatment of breast cancer”. The manuscript ID is pharmaceutics-1613172.

We acknowledge the reviewers for the thoughtful comments on the manuscript. The responses to all comments have been prepared and attached herewith. We have also ensured that our revised files adhere to the author guidelines of your esteemed journal.

Reviewer #3

1- The authors made the proper modification according to the indications and it is now ready to be submitted to the journal for publication.

Response;

We thank reviewer #3 for his thoughtful comments on our manuscript and have edited it to address these concerns. We appreciate your insightful suggestions which have helped to improve the manuscript.

We look forward to working with you and the reviewers to move this manuscript closer to publication in J of Pharmaceutics.

Sincerely,

Shaaban K. Osman

Ahmed A. H. Abdellatif